# Air-stable redox-active nanomagnets with lanthanide spins radical-bridged by a metal–metal bond

Fupin Liu [1], Georgios Velkos[1], Denis S. Krylov[1], Lukas Spree [1], Michal Zalibera[2], Rajyavardhan Ray[1,3], Nataliya A. Samoylova[1], Chia-Hsiang Chen[1], Marco Rosenkranz[1], Sandra Schiemenz[1], Frank Ziegs[1], Konstantin Nenkov[1], Aram Kostanyan[4], Thomas Greber[4], Anja U.B. Wolter[1], Manuel Richter[1,3], Bernd Büchner[1], Stanislav M. Avdoshenko[1] & Alexey A. Popov [1]

Engineering intramolecular exchange interactions between magnetic metal atoms is a ubiquitous strategy for designing molecular magnets. For lanthanides, the localized nature of $4f$ electrons usually results in weak exchange coupling. Mediating magnetic interactions between lanthanide ions via radical bridges is a fruitful strategy towards stronger coupling. In this work we explore the limiting case when the role of a radical bridge is played by a single unpaired electron. We synthesize an array of air-stable $Ln_2@C_{80}(CH_2Ph)$ dimetallofullerenes ($Ln_2 = Y_2$, $Gd_2$, $Tb_2$, $Dy_2$, $Ho_2$, $Er_2$, TbY, TbGd) featuring a covalent lanthanide-lanthanide bond. The lanthanide spins are glued together by very strong exchange interactions between $4f$ moments and a single electron residing on the metal–metal bonding orbital. $Tb_2@C_{80}(CH_2Ph)$ shows a gigantic coercivity of 8.2 Tesla at 5 K and a high 100-s blocking temperature of magnetization of 25.2 K. The Ln-Ln bonding orbital in $Ln_2@C_{80}(CH_2Ph)$ is redox active, enabling electrochemical tuning of the magnetism.

[1] Leibniz Institute for Solid State and Materials Research (IFW Dresden), Helmholtzstrasse 20, 01069 Dresden, Germany. [2] Institute of Physical Chemistry and Chemical Physics, Slovak University of Technology, Radlinského 9, 81237 Bratislava, Slovakia. [3] Dresden Center for Computational Materials Science (DCMS), TU Dresden, D-01062 Dresden, Germany. [4] Physik-Institut der Universität Zürich, Winterthurerstrasse 190, CH-8057 Zürich, Switzerland. These authors contributed equally: Georgios Velkos, Denis S. Krylov, Lukas Spree. Correspondence and requests for materials should be addressed to F.L. (email: f.liu@ifw-dresden.de) or to S.M.A. (email: s.avdoshenko@ifw-dresden.de) or to A.A.P. (email: a.popov@ifw-dresden.de)

anthanides are well-known for their large atomic moments and magnetic anisotropies, and embedding discrete lanthanide ions in molecular environments leads to nanomagnets exhibiting magnetic bistability and slow relaxation of magnetization on a single molecule level[1,2]. Single molecule magnets (SMMs) can be used as core elements of nanospintronic devices[3], such as spin valves[4], spin transistors[5,6], or building blocks of quantum computers[7,8]. Optimization of symmetry and ligand environment led to a dramatic improvement of lanthanide SMMs during the last decade[9–11] with the latest discovery of magnetic hysteresis at 60–80 K in Dy-metallocenium salts[12–15]. Effective barriers of magnetization reversal higher than 1000 K were reported for several lanthanide SMMs[12–20]. Single lanthanide atoms were also found to keep magnetic bistability up to 45 K on MgO|Ag(100) substrate[21–23].

Combination of several magnetic centers within one molecule may lead to high-spin ground states and can largely suppress quantum tunneling, which is the main low-temperature relaxation mechanism for single-ion magnets in zero magnetic field. Design of multinuclear SMMs has been a viable strategy since the discovery of SMM behavior in the $Mn_{12}$ complex[24]. The temperature range at which a multinuclear magnet can be considered as a giant spin rather than a combination of weakly interacting individual spins is limited by the strength of exchange interactions. Whereas exchange interactions between transition metals can be tuned in a wide range, the localized nature of $4f$ electrons results in weak exchange interactions in lanthanide compounds rarely exceeding $1\,cm^{-1}$. As a result, when the relaxation of magnetization in multinuclear lanthanide SMMs is driven via exchange excitations, the barriers to magnetization reversal are usually well below 100 K[25].

Exchange coupling in lanthanide molecule magnets can be increased by introducing radical bridges[26]. The radical bridge usually features a rather diffuse singly occupied molecular orbital, which exhibits stronger interactions with the $4f$ electrons. The lanthanide-radical exchange coupling constants can reach values of $-27\,cm^{-1}$. The strongest coupling so far has been found in dilanthanide complexes with $N_2^{3-}$ radical bridges[27–29], and the corresponding Tb complex has a blocking temperature near 30 K, which is the highest known value among multinuclear SMMs[30]. In this work we explore the limiting case of this concept in which the role of a radical bridge is played by a single unpaired electron, residing on the lanthanide-lanthanide bonding orbital and coupling the lanthanide spins inside a fullerene.

The empty space inside carbon cages provides unmatched possibilities for stabilizing small metallic clusters in unconventional valence and spin states[31], such as metal dimers in dimetallofullerenes. Although Coulomb repulsion between two metal ions prevails over the covalent bonding[32–34], the metal dimers cannot dissociate inside fullerenes. This unique situation allows for direct Ln–Ln bonding in dimetallofullerenes[35–38], which could not be realized in any other molecular lanthanide compound so far. Of particular interest are $Ln_2@C_{80}$ molecules. The $C_{80}$-$I_h$ fullerene cage is unstable in the neutral state due to the presence of a four-fold degenerate orbital occupied by only two electrons. At the same time, the hexaanion of $C_{80}$-$I_h$ has a very stable closed-shell electronic structure, and hence this fullerene is the most preferable host for endohedral species acting as donors of six electrons, such as the early lanthanide dimers $La_2$ or $Ce_2$[39]. However, in the $Ln_2@C_{80}$ molecules with heavier lanthanides (Gd to Lu), the $Ln_2$ dimers transfer only 5 electrons to the fullerene orbitals, leaving one electron on the Ln–Ln bonding orbital[40]. The electronic structure of the fullerene cage in such $Ln_2^{5+}@C_{80}^{5-}$ dimetallofullerenes can be stabilized by addition of one electron[41], substitution of one carbon by a nitrogen atom (giving azafullerenes $Ln_2@C_{79}N$[37,42]) or by functionalization with a radical group (giving derivatives $Ln_2@C_{80}R$, $R = CF_3$[40,43], $CH_2Ph$[38]). A formal oxidation state of the lanthanides in such dimetallofullerenes is $Ln^{+2.5}$. EPR studies of $Gd_2@C_{79}N$ revealed a ferromagnetic ground state with a giant spin of $S = 15/2$, and magnetization studies showed that the exchange coupling between the Gd spin and the unpaired electron residing on the Gd–Gd bonding orbital is as large as $170\,cm^{-1}$ (refs. [44,45]). Very recently we have reported that $Dy_2@C_{80}(CH_2Ph)$ shows exceptional magnetic properties with magnetic hysteresis up to 22 K and Dy-electron exchange coupling of $32\,cm^{-1}$ [38].

In order to investigate and understand the principles underlying the SMM behavior in encapsulated lanthanide dimers, in this work we synthesize and study an array of $Ln_2@C_{80}(CH_2Ph)$ molecules ($Ln_2 = Y_2$, $Gd_2$, $Tb_2$, $Dy_2$, $Ho_2$, $Er_2$, TbY, TbGd), all featuring single-electron Ln–Ln bonding molecular orbitals (MO). This bonding situation leads to giant exchange interactions in all magnetic molecules and is very beneficial for the molecular magnetism, especially in $Tb_2@C_{80}(CH_2Ph)$ showing a giant coercivity and the highest known blocking temperature among dinuclear lanthanide complexes. Furthermore, we demonstrate that the single-electron Ln–Ln bonding MO is redox-active, which allows control of the magnetism via electron transfer.

## Results

**Synthesis and internal dynamics.** All compounds were synthesized using the extraction/functionalization procedure recently developed in our group (Supplementary Fig. 1)[38]. Carbon soot obtained by arc-discharge of metal-oxide filled graphite rods is extracted with hot DMF, giving a mixture of fullerene anions in DMF solution. The anions are reacted with benzyl-bromide to yield neutral air-stable benzyl monoadducts, which are further separated by multistep HPLC protocol to yield pure $Ln_2@C_{80}(CH_2Ph)$ derivatives (denoted as {$Ln_2$} hereafter, Ln = Y, Gd, Tb, Dy, Ho, Er; Supplementary Figs 2–20). If a mixture of two metal oxides (Ln and Ln′) is used in the arc-discharge synthesis, the same procedure gives a mixture of {$Ln_2$}, {$Ln'_2$}, and {LnLn′}. For the Tb-Y mixed-metal system, the mixture is further separated by recycling HPLC to give pure {$Y_2$}, {$Tb_2$} and {TbY}. In the Tb-Gd system, the separation into individual components with HPLC could not be achieved, and the studied sample comprised ca 20% {$Gd_2$}, 30% {$Tb_2$} and 50% {TbGd}. Despite the unconventional oxidation state of the lanthanides (+2.5) and the presence of an unpaired valence electron (Fig. 1), the {$Ln_2$} compounds are air-stable at room temperature (Supplementary Fig. 21) and do not require special handling

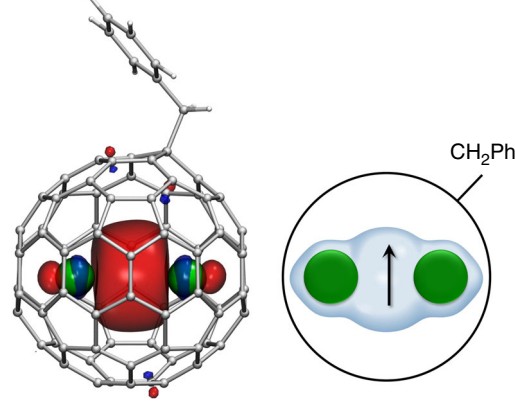

**Fig. 1** Molecular structure of $Ln_2@C_{80}(CH_2Ph)$. Single-occupied Ln–Ln bonding molecular orbital (left; carbons are gray, hydrogens are white, lanthanides are green), and schematic depiction of the molecule (right; the arrow indicates an unpaired electron residing on the Ln–Ln bonding orbital)

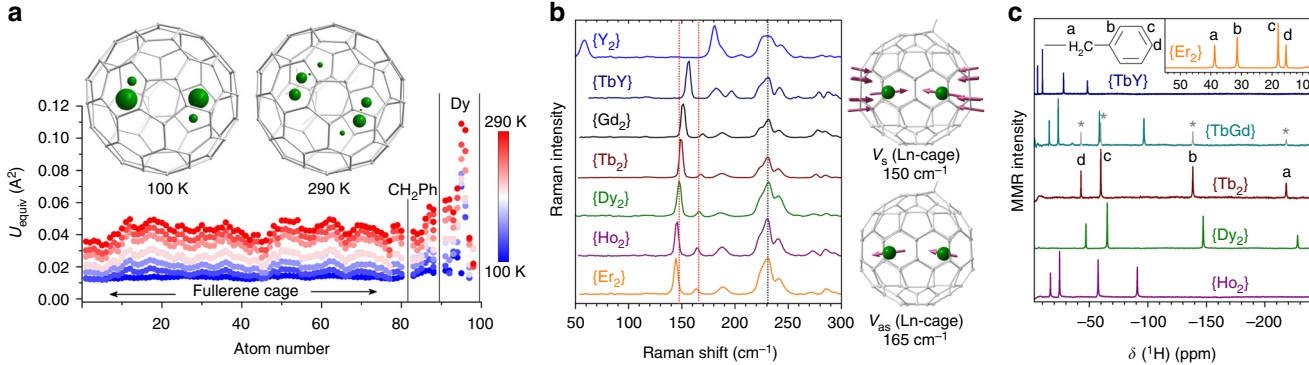

**Fig. 2** Structure and dynamics of $Ln_2@C_{80}(CH_2Ph)$. **a** Molecular structure of {$Dy_2$} at 100 and 290 K and atomic displacement parameters as a function of temperature between 100 and 290 K (to guide the eye, vertical lines separate displacement parameters of the $C_{80}$ cage, $CH_2Ph$ group, and Dy atoms); Dy atoms are shown as spheres with radii proportional to the site occupancies; **b** Raman spectra of {$Ln_2$} compounds in the low-frequency range measured at 77 K; two metal-cage stretching modes are indicated by red dotted lines, the most prominent fullerene cage squashing mode is indicated by a black dotted line, atomic displacements for the metal-cage stretching modes near 150 and 165 cm$^{-1}$ are also shown on the right; **c** $^1$H NMR spectra of {$Ln_2$} measured in $CS_2$ solution at room temperature; {$Tb_2$} signals in the spectrum of the {TbGd} sample are denoted by asterisks

conditions as many Ln compounds in unconventional oxidation states do.

Molecular structure and internal dynamics of the metal atoms inside the fullerene cage are studied by variable-temperature single-crystal X-ray diffraction for {$Dy_2$} as a representative example providing single crystals of sufficient quality (Supplementary Figs 22, 23, Supplementary Table 1). Virtually identical Vis-NIR, IR, and Raman spectra prove that all isolated {$Ln_2$} compounds are isostructural (Supplementary Figs 24–26). Figure 2a shows the molecular structure of {$Dy_2$} at 100 and 290 K along with the temperature dependence of atomic displacement parameters. Whereas the fullerene cage with the attached benzyl group remains ordered at all temperatures, the metal atoms show a pronounced increase of their mobility with temperature. At 100 K each Dy atom occupies two well-defined positions with 70 and 30% occupancy. The increase of temperature to 290 K induces vigorous motions of the metal atoms as can be seen from the increase of the number of metal sites and increase of their displacement parameters compared to those of the carbon atoms (Fig. 2a, Supplementary Table 2, Supplementary Note 1). The disordered metal positions tend to be distributed within a single plain, which suggests that the $Ln_2$ dimer in {$Ln_2$} exhibits in-plain rotation near room temperature.

Translational motions of the metal atoms inside the cage manifest in low-frequency vibrations in the Raman spectra (Fig. 2b). A strong Raman line near 150 cm$^{-1}$ is due to the in-phase cage-metal stretching mode in {$Ln_2$}. Its frequency exhibits pronounced metal dependence and shifts from 144.5 cm$^{-1}$ in {$Er_2$} to 151.2 cm$^{-1}$ in {$Gd_2$} and further to 181.1 cm$^{-1}$ in {$Y_2$} in accordance with the decrease of the metal atomic mass. Similar metal dependence is found for the anti-symmetric metal-cage mode observed as a low-intensity peak shifting from 163.7 cm$^{-1}$ in {$Er_2$} to 169.6 cm$^{-1}$ in {$Gd_2$}. At higher frequencies only carbon atoms contribute to the vibrational displacements, and the IR and Raman spectra of all studied {$Ln_2$} compounds are virtually identical (Supplementary Figs 25, 26).

**Paramagnetic $^1$H NMR spectroscopy.** The $Ln_2$ dimers inside the fullerene cages act as magnets, creating dipolar magnetic fields. The strength and spatial distribution of these fields can be evaluated by $^1$H NMR spectroscopy using benzyl protons as a probe. In solution $^1$H NMR spectra, all {$Ln_2$} compounds except for {$Y_2$} and {$Gd_2$} exhibit well-defined temperature-dependent $^1$H resonances (Fig. 2c, Supplementary Figs 27, 33) strongly shifted from

the standard chemical shifts of the benzyl group in diamagnetic compounds (3–7 ppm). These paramagnetic shifts are caused by the screening of the external magnetic field by the dipolar field of the endohedral lanthanide dimer. Since the molecules in solution are rotating fast, the isotropic contributions average out and the paramagnetic shift ($\delta^{para}$) serves as a measure of a magnetic anisotropy, taking the following form in a point dipole approximation:[46]

$$\delta_i^{para} = \frac{(3\cos^2\theta_i - 1)}{12\pi R_i^3}\left(\chi_\parallel^{Ln_2} - \chi_\perp^{Ln_2}\right) \qquad (1)$$

where the first term is defined via polar coordinates $R_i$ and $\theta_i$ of the $i$-th proton in the coordinate system centered on the $Ln_2$ dimer with polar axis along the Ln–Ln bond and is expected to be very similar for all {$Ln_2$} compounds. The second term in Eq. 1 is the difference of the longitudinal $\chi_\parallel$ and transverse $\chi_\perp$ magnetic susceptibilities of the $[Ln^{3+}-e-Ln^{3+}]$ system. {$Dy_2$} and {$Tb_2$} exhibit almost identical $^1$H chemical shifts, indicating a similarity of their magnetic properties, whereas the values of {$Ho_2$} are at least twice smaller. Substitution of one Tb in {$Tb_2$} by Y results in a 4.3-fold drop of the paramagnetic shift in {TbY}. Remarkably, $^1$H resonances in {$Er_2$} are shifted into the positive direction, revealing that the sign of the magnetic anisotropy in the endohedral $Er_2$ dimer is opposite to that in $Tb_2$, $Dy_2$, and $Ho_2$ dimers.

The isotropic spin of Gd does not induce dipolar shifts in the NMR spectra (although it does affect the relaxation rates of proton spins and thus makes the lines very broad—we, therefore, could not detect the $^1$H NMR spectrum of {$Gd_2$}). If the Gd spin in {TbGd} were behaving isotropically, the $^1$H chemical shifts of {TbGd} would be close to those of {TbY}. However, the measured $^1$H shifts in {TbGd} are two times larger than in {TbY} (Fig. 2c). This shows that the Gd spin is locked to the anisotropic Tb spin by exchange interaction through the unpaired electron spin. Observation of this phenomenon at room temperature indicates that the exchange coupling is very strong.

**Coupling of lanthanide spins in {$Ln_2$}.** The combination of large anisotropic Ln spins with a strong exchange coupling via a delocalized unpaired electron in {$Ln_2$} molecules is promising for SMM. The study of an array of {$Ln_2$} molecules with different lanthanides enables disentanglement of the exchange and anisotropy factors in determining magnetic properties of the {$Ln_2$} system. This task, however, requires a common theoretical framework, which is outlined in this section.

The effective spin Hamiltonian of the $\{Ln_2\}$ molecule includes single-ion ligand-field (LF) effects together with the Kondo description of magnetic interactions between the lowest $J$-multiplet of lanthanides with the partially delocalized unpaired electron occupying the spd-hybrid Ln–Ln bonding orbital:

$$\hat{H}_{spin} = \hat{H}_{LF_1} + \hat{H}_{LF_2} + \hat{H}_{sf}, \qquad (2)$$

where $\hat{H}_{LF_i}$ is the single-ion LF Hamiltonian of the $i$-th lanthanide site. Later, the LF Hamiltonian will be limited to the crystal-field shape commonly used for 4f systems. Further, $\hat{H}_{sf}$ is the spin-fermion Hamiltonian describing direct exchange interactions between the lanthanide ion moments, the kinetic energy for the electron, and an on-site exchange interaction:

$$\hat{H}_{sf} = -2j_{12}\hat{J}_{Ln_1}\hat{J}_{Ln_2} + t\sum_{\sigma}\left[c^{\dagger}_{1\sigma}c_{2\sigma} + c^{\dagger}_{2\sigma}c_{1\sigma}\right] - 2\hat{s}(K_1\hat{J}_{Ln_1} + K_2\hat{J}_{Ln_2}).$$
$$\qquad (3)$$

Here $j_{12}$ is the direct exchange coupling between the localized lanthanide moments $\hat{J}_{Ln_i}$, $t$ is the electron hopping amplitude between sites 1 and 2, $c_{i\sigma}$ ($c^{\dagger}_{i\sigma}$) is the creation (annihilation) operator for the electron at site $i$ with spin $\sigma$, and $K_i$ denotes the on-site Kondo exchange coupling constant between the localized $4f$ moment $\hat{J}_{Ln_i}$ and the delocalized spin $\hat{s}$. In the modelling of magnetic properties of $\{Ln_2\}$ molecules using spin Hamiltonians derived from Eqs. 2 and 3, lanthanide moments $\hat{J}_{Ln_i}$ are treated in a full $|J, m_J\rangle$ basis set for each lanthanide ion. In general, spin-spin interactions in Eq. 3 are anisotropic and may lead to a rich phase diagram in the full parameter space due to an interplay of very different energy scales[47,48]. A complete derivation of all the parameters is beyond the scope of the present report[49,50]. Nevertheless, this spin Hamiltonian can be simplified by considering the symmetry and properties of individual $\{Ln_2\}$ systems.

**Magnetic properties of $\{Gd_2\}$.** Magnetization curves of $\{Gd_2\}$ show no hysteresis down to 1.8 K, and we did not succeed in determining relaxation of magnetization by using AC magnetometry. Thus, $\{Gd_2\}$ is not a SMM, which is not very surprising since $Gd^{3+}$ ions are magnetically isotropic. Hence, $\{Gd_2\}$ provides a convenient example to study the role of the exchange interactions without the contribution of any single-ion anisotropy.

A recent theoretical study of the $[Gd_2@C_{80}]^-$ anion showed that the hopping amplitude $t$ exceeds $10,000\ cm^{-1}$ (ref. 51), which suggests a weak Kondo coupling limit ($K/t \ll 1$). At this limit, the kinetic term in Eq. 3 can be omitted, and the form of the spin Hamiltonian can be simplified to a simple 3-center model with only exchange interactions between the lanthanides and the electron spin[30,44,45,49,52–54]. As the ligand-field terms for isotropic Gd spins can be neglected as well, the effective spin Hamiltonian of $\{Gd_2\}$ takes the form:

$$\hat{H}_{spin}(\{Gd_2\}) = -2j_{12}\hat{S}_{Gd_1}\hat{S}_{Gd_2} - 2\hat{s}\left(K_1\hat{S}_{Gd_1} + K_2\hat{S}_{Gd_2}\right)$$
$$\approx -2K^{eff}\hat{s}(\hat{S}_{Gd_1} + \hat{S}_{Gd_2}), \qquad (4)$$

where $\hat{S}_{Gd_i}$ denotes the Gd spin at site $i$. Density-functional theory (DFT) calculations predict two very close $K_i$ values for $\{Gd_2\}$, 181 and 184 $cm^{-1}$, and much smaller direct Gd–Gd exchange coupling, $j_{12} = -1.2\ cm^{-1}$.[38] When $j_{12}$ is that small, the $K_i$ and $j_{12}$ parameters cannot be determined from experimental data separately, leading to the approximate Hamiltonian Eq. 4 with an effective coupling $K^{eff}$ (ref. 45). Having used this single-parameter Hamiltonian in the analysis of the measured magnetization and $\chi T$ curves for $\{Gd_2\}$ we have obtained the $K^{eff}$ value of $160 \pm 10\ cm^{-1}$ (Supplementary Figs 34, 36) which is close to the one of the $[Gd^3$

$+-e-Gd^{3+}]$ spin-system in $Gd_2@C_{79}N$ ($K^{eff} = 170 \pm 10\ cm^{-1}$)[45] and is in good agreement with the DFT results.

Strong exchange interactions result in the giant-spin magnetic ground state of $\{Gd_2\}$ with $S = 15/2$, created by two local Gd spins ($S_{Gd} = 7/2$) ferromagnetically coupled via the free electron spin ($S_e = 1/2$). Fine details of this state can be further attested by EPR spectroscopy. At room temperature in toluene solution, $\{Gd_2\}$ shows a single EPR line with a g-factor of 1.987 (Supplementary Figs 37, 38). Freezing the solution at 100 K results in a complex multiline structure in the X-band (9.4 GHz) EPR spectrum (Fig. 3a). The Q-band (34 GHz) spectrum of $\{Gd_2\}$ measured under similar conditions has a simpler but still complex pattern.

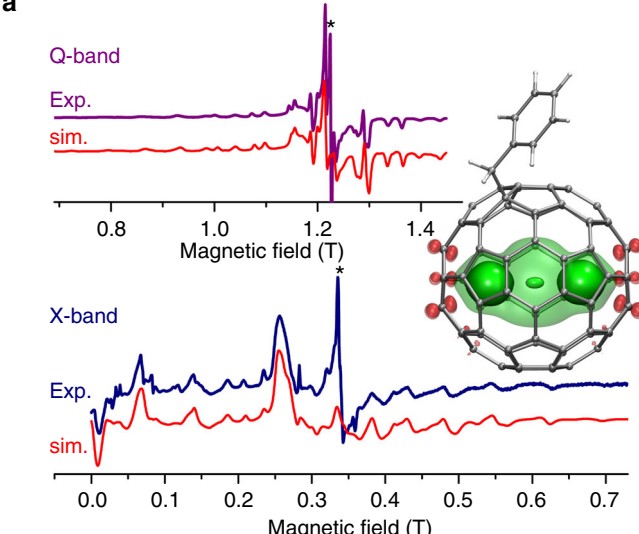

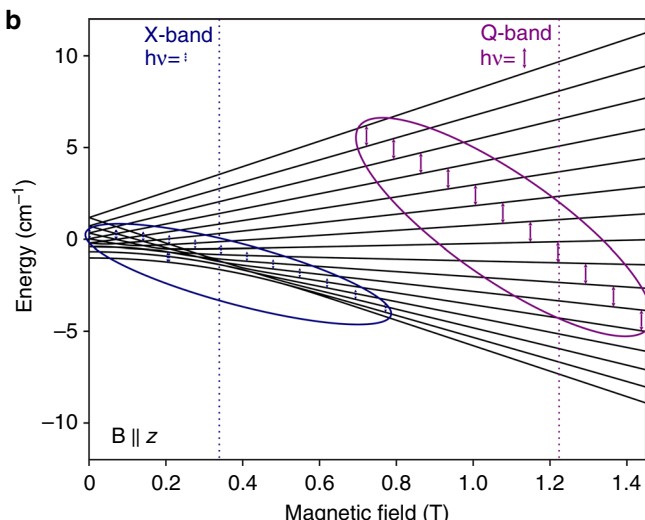

**Fig. 3** Electron paramagnetic resonance (EPR) spectroscopy of $\{Gd_2\}$. **a** X-band and Q-band EPR spectra of frozen $\{Gd_2\}$ solution in toluene near 100 K together with the spectra simulated for spin $S = 15/2$ with $g_{iso} = 1.987$ and zero field splitting (ZFS) parameters $D = 1.00$ GHz and $E = 0.22$ GHz (inhomogeneous broadening is accounted for by ZFS strain $StrD = 0.029$ GHz and $StrE = 0.027$ GHz); asterisks mark unidentified signals (presumably of low spin states or organic impurities), the inset shows the spin-density distribution in $\{Gd_2\}$; **b** Zeeman splitting for spin $S = 15/2$ with the above ZFS parameters (magnetic field is parallel to z-axis of the ZFS tensor); also shown are energies of the X-band (9.4 GHz) and Q-band (34 GHz) microwave photons, EPR-active transitions (ovals and small arrows), and the resonance fields corresponding to the g-factor of 1.987 (vertical dotted lines)

The low-temperature structure in the EPR spectra is an evidence of zero-field interactions in the large-spin ground state. The spin Hamiltonian of such a system can be written as:

$$\hat{H}_{spin} = D\left(\hat{S}_z^2 - \frac{1}{3}S(S+1)\right) + \frac{1}{2}E\left(\hat{S}_+^2 + \hat{S}_-^2\right) + g_{iso}\mu_B\boldsymbol{B}\hat{S}, \quad (5)$$

where the first two terms describe the second-order zero-field splitting (ZFS) of rhombic symmetry, and the last term represents the Zeeman effect. The X-band and Q-band EPR spectra of {Gd₂} in a frozen solution can be well reproduced by the parameters $D = 1.00(2)$ GHz, $E = 0.22(4)$ GHz, and $g_{iso} = 1.987$ (Fig. 3a). The ZFS tensor of {Gd₂} is found to be similar to that of the previously reported $Gd_2@C_{79}N$ ($D = 0.96(6)$ GHz, $E = 0.14(1)$ GHz, $g_{iso} = 1.99$)[44,55], but shows somewhat larger rhombicity, which is in line with the asymmetric geometry of {Gd₂} induced by the exohedral $CH_2Ph$ group. A schematic description of the Zeeman splitting of the 16 energy levels of the weakly anisotropic $S = 15/2$ system in {Gd₂} together with the transitions accessible in the X- and Q-band EPR spectra are shown in Fig. 3b and Supplementary Fig. 39.

**Blocking of magnetization in {Ln₂}.** An essential characteristic of SMM is the blocking of magnetization at a certain temperature (when relaxation of magnetization becomes too slow at the time scale of the measurement). This temperature can be identified by

a characteristic divergence of magnetic susceptibilities $\chi_{FC}$ and $\chi_{ZFC}$, measured for a field-cooled and a zero-field cooled sample, respectively. The blocking temperature of magnetization, $T_B$, is usually defined as the peak in $\chi_{ZFC}$. Four {Ln₂} compounds exhibit blocking of magnetization above 2 K. The $T_B$ value of {Tb₂}, 28.9 K (Fig. 4a), is one of the highest among all known SMMs[12,18,30,56–58]. {Dy₂} also features a high $T_B$ value of 21.9 K[38]. The $\chi_{ZFC}$ of the mixed Tb-Gd sample shows two peaks at 14.4 K and near 29 K (Fig. 4a). The latter corresponds to {Tb₂}, whereas the peak at 14.4 K can be assigned to {TbGd} since {Gd₂} does not show blocking of magnetization above 1.8 K. Finally, in {TbY} $\chi_{FC}$ and $\chi_{ZFC}$ diverge below 5 K, although there is no peak in $\chi_{ZFC}$. Another universal SMM parameter, the 100-s blocking temperature $T_{B100}$, is determined from relaxation times of magnetization (see below) to be 18.2 K for {Dy₂} and 25.2 K for {Tb₂}. $T_{B100}$ of {Tb₂} is surpassed only by a recently reported group of Dy-metallocenium salts with different alkyl groups in cyclopentadienyl rings, which show $T_{B100}$ values of 53–65 K[12–15].

**Magnetic properties of {Tb₂} and {Dy₂}.** In accordance with its high $T_B$, {Tb₂} shows magnetic hysteresis up to a temperature of 28 K (sweep rate 9.5 mT s⁻¹). The hysteresis is extremely broad (Fig. 4b), with giant coercive fields of 8 T at 10 K and 8.2 T at 5 K. This value is similar to that of the recently reported dinuclear Tb-metallocene with $N_2^{3-}$ radical bridge[30] (which also has a high $T_{B100}$ of 20 K) and has no further analogs among molecular

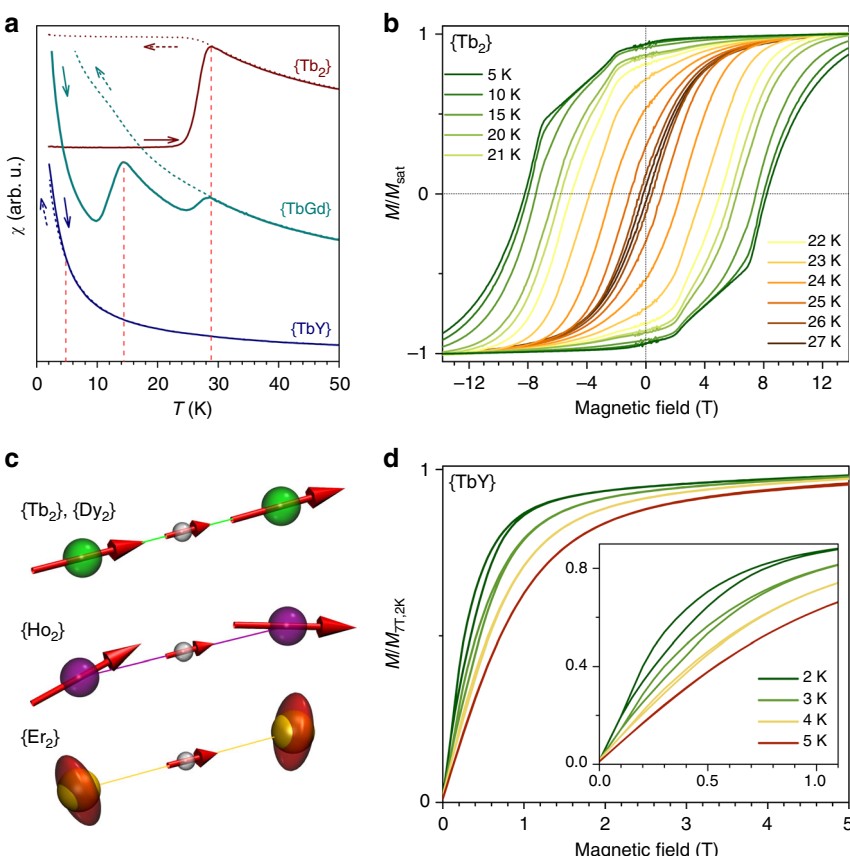

**Fig. 4** Magnetic properties of {Ln₂} molecules. **a** Blocking temperature of magnetization in {Tb₂}, {TbGd} and {TbY}; dotted lines are measurements of magnetic susceptibility $\chi$ during cooling in the field of 0.2 T, solid lines are measurements during heating in the field of 0.2 T of zero-field cooled samples (sweep rate 5 K min⁻¹, arrows indicate direction of the measurement for each curve), vertical red dotted lines denote $T_B$ values; **b** magnetic hysteresis curves for {Tb₂}, sweep rate 9.5 mT s⁻¹; **c** alignment of Ln magnetic moments in {Ln₂} according to ab initio calculations: collinear in {Tb₂} and {Dy₂}, tilted in {Ho₂} (the arrows indicate directions of the single-ion quantization axis for each Ho), easy-plane in {Er₂}, in the latter the Ln spins are visualized as ellipsoids; **d** low-temperature magnetization curves for {TbY}, sweep rate 2.9 mT s⁻¹; the inset shows enhancement of the field range, in which magnetic hysteresis is observed

magnets or bulk magnetic materials. {Dy$_2$} also exhibits magnetic hysteresis up to 21 K with a coercive field of 1.2 T at 1.8 K[38].

The relaxation times of magnetization below $T_B$ are determined by stretched exponential fitting of magnetization decay curves (Supplementary Figs 40, 41). Between 2 and 15 K in zero field {Tb$_2$} exhibits a temperature-independent relaxation time of $(6.5 \pm 1) \times 10^4$ s, which is an indication of the quantum tunneling of magnetization (QTM, Fig. 5a). In this relaxation regime, the giant combined spin of the [Tb$^{3+}$–$e$–Tb$^{3+}$] system flips as a single entity. When QTM is quenched by a finite magnetic field, the relaxation times of {Tb$_2$} are further increased by orders of magnitude. A conservative estimation for the relaxation time in the field of 0.3 T is reaching 6 years at 3 K. Above 20 K the relaxation of magnetization in {Tb$_2$} shows a linear temperature dependence in Arrhenius coordinates. Between 35 and 45 K the relaxation times are determined from AC measurements (Supplementary Fig. 42). The $\tau_m$ values continue the linear regime found in DC measurements, which is well described by the Orbach relaxation mechanism, $\tau_m^{-1} = \tau_0^{-1} \exp(-U^{eff}/T)$, with $U^{eff} = 799 \pm 2$ K and $\tau_0 = (1.66 \pm 0.14) \times 10^{-12}$ s. Here $U^{eff}$ is the effective barrier corresponding to the excited spin state involved in the relaxation, and $\tau_0$ is the attempt time. The $U^{eff}$ value determined for {Tb$_2$} is the largest among all radical-bridged lanthanide molecule magnets described, hitherto.

The relaxation of magnetization in {Dy$_2$} follows the same trend as in {Tb$_2$}, but with lower temperatures and shorter times[38]. A zero-field QTM regime with $\tau_{QTM}$ of $(3.3 \pm 0.2) \times 10^3$ s is found for {Dy$_2$} below 5 K. Above 20 K, relaxation of magnetization in {Dy$_2$} is described by the Orbach mechanism with $U^{eff} = 613 \pm 8$ K and $\tau_0 = (3.6 \pm 1.0) \times 10^{-12}$ s.

The spin Hamiltonian for anisotropic lanthanides requires single-ion ligand field parameters, which were computed here ab initio. At first, the {Ln$_2$} molecules were optimized at the DFT level (Supplementary Tables 3–6), and then the LF states were computed at the CASSCF/RASSI-SO level for each Ln center in a model {LnY}$^-$ system, where Y substitutes one of the lanthanide ions (Supplementary Tables 7–11). The calculations revealed that Tb$^{3+}$ and Dy$^{3+}$ in the {Ln$_2$} molecules have easy-axis anisotropies with a high-spin ground state, which are described by $|\pm 15/2\rangle$ and $|\pm 6\rangle$ doublets for Dy and Tb, resp. (complete details of the LF splitting are described in Supplementary Table 8, possible origins of the axial ligand field imposed on lanthanide ions in {Ln$_2$} molecules are discussed in Supplementary Notes 2 and 3, Supplementary Fig. 43, Supplementary Table 12 and in

ref. 38). Thus, the ground-state spins of Tb or Dy are of Ising type with their easy axes aligned along the Ln–Ln bond. Therefore the low-energy states of the [Ln$^{3+}$–$e$–Ln$^{3+}$] systems can be described by the same effective spin Hamiltonian as in Eq. 4, but with addition of the ligand field terms:

$$\hat{H}_{spin}(\{Ln_2\}) = \hat{H}_{LF_1} + \hat{H}_{LF_2} - 2K^{eff}\hat{s}\left(\hat{J}_{Ln_1} + \hat{J}_{Ln_2}\right). \quad (6)$$

Here again, the direct Ln–Ln exchange is neglected. A similar form of the Hamiltonian can be derived from the Lines model[59], assuming isotropic coupling for each Kramers Doublet (KD). Although a description of the exchange in Eq. 6 as isotropic is an oversimplification, for the Ising ground state with collinear spins this approximation is essentially valid. Thus, the Hamiltonian Eq. 6 provides a correct description of the ground state properties of the systems with strong easy-axis anisotropy, such as {Tb$_2$} and {Dy$_2$}, but it is expected to be less reliable when interactions with higher KDs are involved. Since the relative energies of the first and second excited doublets, KD2 and KD3, are predicted to be in the range of 300–500 cm$^{-1}$ for Tb and 200–400 cm$^{-1}$ for Dy, only the lowest-energy excited states contribute to the magnetic properties in the experimentally relevant temperature range.

The values of $K^{eff}$ for {Tb$_2$} and {Dy$_2$} can be estimated by modeling the susceptibility and the magnetization curves using Hamiltonian Eq. 6 with the addition of the Zeeman term[60]. The shapes of these curves (Supplementary Figs 44–46) correspond to the strong coupling with the $K^{eff}$ values of 45–53 cm$^{-1}$ for {Tb$_2$} and 30–35 cm$^{-1}$ for {Dy$_2$}. For such large values, the estimates from $\chi T$ fitting give rather large confidence limits. Yet, a more precise estimation is possible through modeling of the effective barriers of the Orbach process, $U^{eff}$. It can be done assuming that the low-energy LF excited states do not participate in the relaxation of {Tb$_2$} and {Dy$_2$}, while an efficient Orbach process involves the first exchange-excited state corresponding to flipping of one of the lanthanide spins under the Hamiltonian Eq. 6 (Fig. 5a)[30,38,49]. If only the exchange term of the Hamiltonian Eq. 6 is considered, and the ground state lanthanide spins are of Ising type with $J_z = \pm J$ (here $J$ is the total momentum of the lanthanide ion), the energy of the exchange-excited state and hence the relaxation barrier would be $U^{eff} = 2JK^{eff}$ with $K^{eff}$ of 46 cm$^{-1}$ for {Tb$_2$} and 28 cm$^{-1}$ for {Dy$_2$} (see ref. 30 for a similar discussion on the radical-bridged dinuclear Tb complex). However, mixing of the LF and exchange excitations changes the energies of the states with predominant exchange excitation

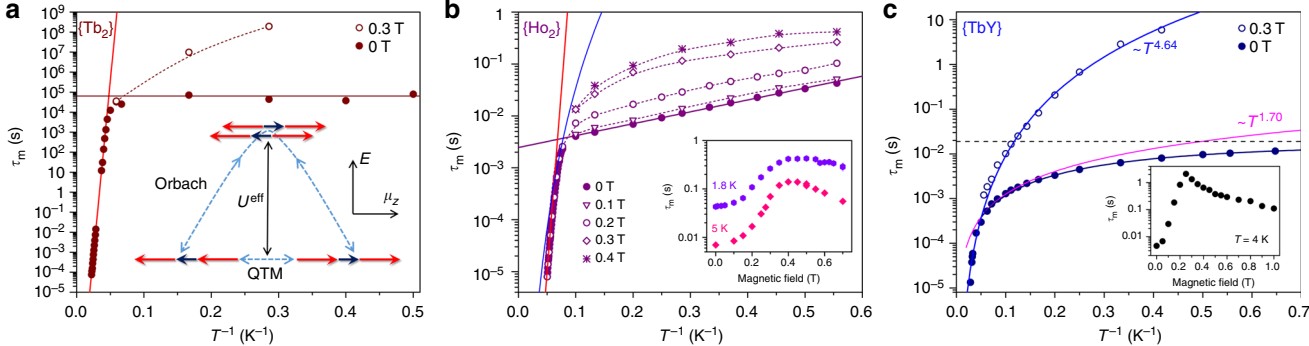

**Fig. 5** Relaxation of magnetization in {Ln$_2$} molecules. **a** magnetic relaxation times of {Tb$_2$}, full dots are zero-field data, open dots are in-field data, red line denotes Orbach processes, solid horizontal line denotes QTM; the inset shows schematically two main relaxation pathways in {Tb$_2$}: QTM and Orbach relaxation via an exchange-excited state, red arrows are Ln spin, blue arrow is a free electron spin; **b** magnetic relaxation times of {Ho$_2$} at zero field (full dots) and in different fields between 0.1 and 0.4 T, red and purple solid lines are Orbach processes, blue line is a possible Raman contribution ($\sim T^{10.1}$); the inset shows magnetic field dependence of relaxation times at 1.8 and 5 K; **c** magnetic relaxation times of {TbY}, dashed horizontal line is a QTM contribution to zero-field relaxation, magenta line is a low-power process ($\sim T^{1.70}$), dark blue line is a combination of both, light blue line is a Raman process ($\sim T^{4.64}$)

(Supplementary Table 13, Supplementary Fig. 47, Supplementary Note 4), resulting in the $K^{eff}$ values of 55 cm$^{-1}$ and 32 cm$^{-1}$ for {Tb$_2$} and {Dy$_2$} systems, respectively. For comparison, the largest lanthanide-radical coupling constants in the radical-bridged complexes are −23.1 cm$^{-1}$ for Tb and −7.2 cm$^{-1}$ for Dy in [Ln$^{3+}$–N$_2$$^{3-}$–Ln$^{3+}$] systems[30].

**Magnetic properties of {Ho$_2$}**. Relaxation times of {Ho$_2$} determined by AC magnetometry between 1.8 and 20 K (Supplementary Figs 48–53) show peculiar temperature and field dependences (Fig. 5b). Between 1.8 and 10 K in zero DC field, the compound exhibits a linear log($\tau_m$)-vs-$T^{-1}$ dependence resembling the Orbach mechanism with $U^{eff} = 5.3 \pm 0.1$ K and $\tau_0 = 2.46 \pm 0.06$ ms. With the increase of the magnetic field up to 0.5 T, the relaxation decelerates, and the linear dependence is gradually transformed into a curved one (Fig. 5b). If the magnetic field exceeds 0.5 T, the relaxation accelerates again. Deceleration of the relaxation of magnetization with the application of a magnetic field is a typical characteristic of QTM, but the pronounced temperature dependence is not common for a ground-state QTM. However, Zheng and Chilton et al. have recently emphasized that QTM may show temperature dependence due to the temperature-dependent phonon collision rate[61], and temperature-dependent QTM was also observed in DySc$_2$N@C$_{80}$[62]. Alternatively, such a behavior may correspond to the thermally-assisted QTM, i.e., QTM in an excited state, but the nature of this state is not clear since the energy of 5.3 K is much smaller than the predicted ligand field splitting (see below). The increase of the relaxation rate with the further increase of the magnetic field is an indication of the relaxation via the direct mechanism, which involves phonons of the frequency corresponding to the energy gap between the opposite spin states.

Above 10 K, the relaxation of magnetization in {Ho$_2$} is field-independent and exhibits rapid acceleration with temperature. The log($\tau_m$)-vs-$T^{-1}$ dependence deviates from the linear form and cannot be assigned to a single Orbach process (Fig. 5b). The best fit to the data is obtained by a combination of Orbach and Raman mechanisms (the latter implies a relaxation rate as power function of temperature, $\tau_m^{-1} = AT^n$) with $U^{eff} = 334 \pm 10$ K, $\tau_0 = (5.6 \pm 2.6) \times 10^{-13}$ s, $n = 10.1 \pm 0.3$, and $A = (1.71 \pm 1.3)$ $10^{-9}$ s$^{-1}$ K$^{-10.1}$. A similarly good fit is obtained for a combination of two Orbach processes with parameters $U_1^{eff} = 324 \pm 5$ K, $\tau_{01} = (0.8 \pm 0.2) \times 10^{-12}$ s, and $U_2^{eff} = 136 \pm 5$ K, $\tau_{02} = (1.1 \pm 0.4) \times 10^{-7}$ s.

Ab initio calculations for Ho$^{3+}$ in {Ho$_2$} predict a high-spin ground state and smaller energy splitting compared to {Tb$_2$} and {Dy$_2$}. Further, the easy-axis of each ion is tilted from the Ho–Ho axis by 13.4° (Fig. 4c), and the quasi-doublet ground states have strongly mixed $m_J$ character due to higher-order LF terms (Supplementary Tables 9 and 12, Supplementary Note 3): the leading $m_J$ terms are 64% $|\pm 8\rangle$, 14% $|\pm 7\rangle$, and 10% $|\pm 6\rangle$. The lower energy splitting and non-collinearity of the single-ion magnetic moments are consistent with the smaller paramagnetic shifts observed for {Ho$_2$} in the $^1$H NMR spectra when compared to {Tb$_2$} and {Dy$_2$}.

For non-collinear magnetic moments of Ho ions in {Ho$_2$} the anisotropy of the exchange coupling should have a pronounced effect already in the ground state, and the spin Hamiltonian Eq. 6 cannot capture the whole underlying physics. Indeed, our attempts to reproduce the experimental $\chi T$ and magnetization curves of {Ho$_2$} using Hamiltonian Eq. 6 gave poorer agreement than for {Tb$_2$} and {Dy$_2$} (Supplementary Figs 54–56). The closest match between experiment and simulations is found for $K^{eff} = 40$ cm$^{-1}$. With this value of the coupling constant, the lowest-energy exchange-excited state is predicted at 374 K (Supplementary Table 14, Supplementary Fig. 57), which can be compared to the estimated $U^{eff}$ value of 324–334 K.

The non-collinearity effect can alternatively be introduced via phenomenological Dzyaloshinkii-Moriya interaction term between the lanthanide sites, $\hat{H}_{DMI} = \mathbf{D}_{12} \cdot (\hat{J}_{Ln_1} \times \hat{J}_{Ln_2})$, where $\mathbf{D}_{12}$ is the Dzyaloshinskii–Moriya vector[63]. Presence of this term leads to staggered magnetization and also allows for the exchange and Kondo spin-fluctuation processes, providing a spin relaxation mechanism for {Ho$_2$}.

**Magnetic properties of {Er$_2$}**. For {Er$_2$}, dynamic susceptibility measurements showed the out-of-phase $\chi$" response only at low temperature and in the presence of a magnetic field (Supplementary Figs 58–60). At 1.8 K, the relaxation time increases with the field from 18 ms at 0.1 T to 47 ms near 0.5 T, but the amplitude of $\chi$" is the highest in the field of 0.25 T. The temperature dependence measured in the field of 0.25 T revealed a decay of the relaxation time from 30 ms at 1.8 K to 10 ms near 3 K; at higher temperatures the signal becomes too weak to be measured reliably. The assignment of the underlying relaxation process to the SMM features of {Er$_2$} is ambiguous because similar relaxation behavior at low temperatures may be also caused by the lattice-based phonon bottleneck. In good agreement with paramagnetic NMR data, our ab initio calculations predict an easy-plane character of the ground state of Er$^{3+}$ ions in {Er$_2$} as visualized in Fig. 4c (see also Supplementary Table 10 and extended discussion in Supplementary Note 3). In this situation the simple spin Hamiltonian Eq. 6 cannot describe the spin exchange processes well (Supplementary Figs 61–63). Moreover, the easy-plane anisotropy implies the presence of strong spin-fluctuation processes, which provides spin relaxation mechanism detrimental to good SMM behavior.

**Magnetic properties of {TbY}**. To understand, if the symmetry of the [Ln$^{3+}$–$e$–Ln$^{3+}$] spin system is essential for the excellent SMM performance of {Tb$_2$}, we studied mixed-metal {TbGd} and {TbY} compounds. The blocking of magnetization near 14 K in {TbGd} shows that coupling the large isotropic spin of Gd to the anisotropic spin of Tb via the delocalized electron spin gives reasonably strong SMM, yet the absence of anisotropy on one of the metal sites leads to the two-fold decrease of the blocking temperature.

Substitution of one Tb ion in {Tb$_2$} by a non-magnetic Y results in a dramatic increase of the relaxation rate. {TbY} shows narrow magnetic hysteresis only below 5 K (Fig. 4d, Supplementary Figs 64, 65). The opening is observed in the field range of 0.1–1.0 T, whereas near zero field the hysteresis loop is closing. AC measurements (Supplementary Figs 66–68) also showed that in-field and zero-field magnetization relaxation times of {TbY} are considerably different below 15 K (Fig. 5c), and the difference is reaching a factor of 450 at 2 K (2.9 s at 0.3 T versus 6 ms in zero field). The field dependence of $\tau_m$ measured at 4 K has a sharp maximum at 0.25 T. Such a strong variation of relaxation time with the magnetic field points to a considerable contribution of zero-field QTM at low temperature. However, the zero-field relaxation rate shows temperature dependence down to 1.8 K. The low-$T$ part can be well described by a combination of temperature-independent QTM and a power function of temperature, $\tau_m^{-1} = \tau_{QTM}^{-1} + AT^n$, with $\tau_{QTM} = 19.0 \pm 0.6$ ms, $A = 16 \pm 1$ s$^{-1}$ K$^{-n}$, and $n = 1.70 \pm 0.04$. The exponent of 1.7 is close to the expected value for a direct ($n = 1$) or a bottlenecked direct process ($n = 2$). However, this temperature-dependent process should be strongly linked to the QTM because it is not observed anymore when the finite field of 0.3 T is applied. Temperature dependence of the in-field relaxation rate as well as

high-temperature zero-field relaxation are well described by a power function with parameters $A = 2.5 \pm 0.5\,\mathrm{ms}^{-1}\,\mathrm{K}^{-n}$, and $n = 4.64 \pm 0.08$. Fitting of $\chi T$ and magnetization measurements with the short version of the Hamiltonian Eq. 6, including only single lanthanide ion exchange-coupled to electron spin gives the optimal $K^{\mathrm{eff}}$ value of $35\,\mathrm{cm}^{-1}$ (Supplementary Figs 69–71), which is considerably smaller than the Tb-electron coupling constant in {Tb$_2$}. These results prove that the coupling of the single lanthanide spin to a delocalized electron spin of the single-electron Tb–Y bond is not sufficient to create a strong SMM and that the presence of two local lanthanide spins in {Ln$_2$}, preferably both of uniaxial anisotropy type, is indeed essential.

**Electrochemistry and properties of {Ln$_2$}$^-$ anions.** The Ln–Ln bonding orbital occupied by a single electron is expected to be redox-active. All {Ln$_2$} compounds exhibit reversible electrochemistry (Fig. 6a shows the cyclic voltammogram of {Er$_2$} as a representative example, see Supplementary Figs 72–77 for other {Ln$_2$} molecules) with almost identical first oxidation potentials at $0.50-0.52\,\mathrm{V}$ and strongly metal-dependent first reduction potentials varying from $-0.86\,\mathrm{V}$ in {Gd$_2$} to $-0.42\,\mathrm{V}$ in {Er$_2$} (Fig. 6, potentials are referred versus Fe(Cp)$_2^{+/0}$ redox couple and listed in Supplementary Table 15). The first reduction potentials correlate well with Shannon ionic radii[64] of the metals; even better correlation is found between the first reduction potentials and the $4f^n5d^16s^2 \rightarrow 4f^n5d^26s^1$ excitation energies of lanthanide atoms (Supplementary Fig. 78). The metal dependence is a clear indication of the population of the single-electron Ln–Ln bonding orbital by the second electron in the {Ln$_2$}$^-$ anion as sketched in

Fig. 6b. The formal Ln$^{2.5+}$ oxidation state of the pristine {Ln$_2$} is thus transformed into the Ln$^{2+}$ state in {Ln$_2$}$^-$.

Transformation of the single-electron Ln–Ln bond into the two-electron covalent bond upon the first reduction should decrease the exchange coupling between the lanthanide spins. The system of two weakly-coupled spins in {Ln$_2$}$^-$ would have a much smaller magnetic anisotropy and hence a smaller paramagnetic shift of the $^1$H NMR resonances than the strongly coupled dimer in {Ln$_2$}. Indeed, $^1$H NMR spectra of the anions {Tb$_2$}$^-$ and {Ho$_2$}$^-$, produced by a single-electron reduction of pristine {Ln$_2$} compounds with cobaltocene, show a considerable decrease of the $^1$H paramagnetic shifts (Fig. 6d). Surprisingly, in {Er$_2$}$^-$ the dipolar shifts are increased when compared to the pristine {Er$_2$}. The latter indicates that the anisotropic exchange interactions in {Er$_2$} are enhancing the axial ($z$) component of the magnetic susceptibility and hence decrease the total magnetic anisotropy in comparison to the weakly-coupled spin system in {Er$_2$}$^-$.

**Discussion**

To summarize, we showed that just like delocalization of electrons is the essence of covalent chemical bonding, delocalization of the unpaired electron spin in the [Ln$^{3+}$–$e$–Ln$^{3+}$] system between two lanthanide sites glues their magnetic moments together. The strong ferromagnetic coupling emerging from these interactions is responsible for the high-spin magnetic ground state in certain {Ln$_2$} molecules. But the strong exchange coupling alone is not enough to make a good SMM. Single-ion anisotropy and collinearity of lanthanide spins play a crucial role as well, and whereas {Tb$_2$} and {Dy$_2$} with high-spin easy-axis single-ion

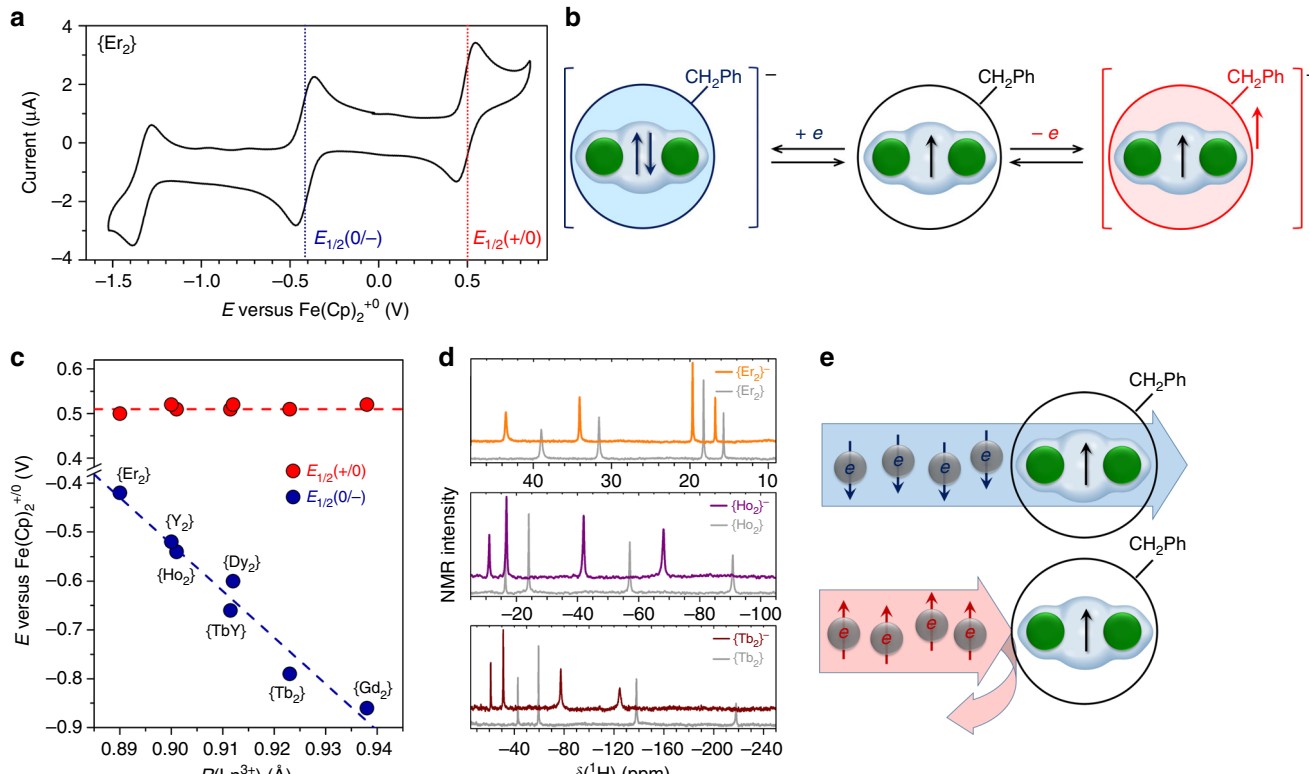

**Fig. 6** Electron transfer properties of {Ln$_2$} molecules. **a** Cyclic voltammogram of {Er$_2$} in $o$-dichlorobenzene solution as a representative example of the {Ln$_2$} series. **b** Schematic description of the single-electron reduction and oxidation of {Ln$_2$} compounds showing addition of one electron to the Ln–Ln bond and removal of one electron from the fullerene cage. **c** The first oxidation (red dots) and reduction (blue dots) potentials of {Ln$_2$} in $o$-dichlorobenzene/TBABF$_4$ solution as a function of ionic radius of Ln (for {TbY}, the average radius of Tb$^{3+}$ and Y$^{3+}$ is used; lines are shown to guide the eye; **d** $^1$H NMR spectra of {Tb$_2$}$^-$, {Ho$_2$}$^-$ and {Er$_2$}$^-$ anions in $d_4$-$o$-dichlorobenzene (colored lines) in comparison to the spectra of neutral compounds (light gray lines). **e** Schematic description of the spin-valve effect of the {Ln$_2$} molecule: in a certain bias range limiting the current to the metal-based LUMO, only the electrons with their spin antiparallel to the spin of the molecule can pass through

ground states and collinear moments make the best SMMs in the series, {Ho$_2$} with mixed LF states and tilted magnetic moments is only a modest SMM, and {Er$_2$} with easy-plane single-ion anisotropy is hardly an SMM at all. The different magnetic anisotropy of these chemically very similar molecules can be understood from the element-specific shape of the Ln-4$f$ charge density, interacting with the molecular charge distribution and being spin-orbit coupled to the 4$f$ magnetic moments (Supplementary Note 3). Furthermore, the coupling of a single lanthanide spin to the delocalized electron spin is also not sufficient as illustrated by the soft SMM behavior of the mixed-metal {TbY} system. Thus, homonuclear lanthanide dimers with collinear magnetic moments and strongly-axial single-ion magnetic anisotropy give the best SMMs.

Importantly, although the Ln$_2$ dimer is protected by the fullerene, it is not completely isolated from the environment. The carbon cage remains transparent for electrons[65,66], and {Ln$_2$} compounds exhibit lanthanide-based redox-activity. In the first reduction step, the Ln–Ln bonding orbital is populated by a second electron, thus allowing to change the valence state from Ln$^{+2.5}$ to Ln$^{+2}$. Simultaneously, the exchange interactions between Ln spins are reduced in the anionic state. Importantly, the valence electrons are strongly coupled to the lanthanide spins. Thus, magnetic properties of {Ln$_2$} molecules can be switched by an electron transfer, which forms a background for their possible application in spin-polarized molecular transport, as redox magnetic switches or as an electron spin detector using magnetoresistance (Fig. 6e).

## Methods

**Synthesis**. {Ln$_2$} compounds were produced by the Krätschmer-Huffman method followed by functionalization with a benzyl group using the method developed in ref. 38 (Supplementary Fig. 1) The graphite rods (length 100 mm, diameter 6 mm) are packed with metal oxides mixed with graphite (molar ratio of Ln:C = 1:15) and evaporated in an electric arc with a current of 100 A in 180 mbar helium atmosphere. The fullerene-containing soot is extracted under nitrogen for 20 h by boiling dimethylformamide (DMF), and DMF solution of EMFs is then reacted with excess of benzyl bromide BrCH$_2$Ph for 20 h at elevated temperature under nitrogen protection. Afterwards the solvent is evaporated, and the residue is washed with methanol. The rest is dissolved in toluene and further separated by high performance liquid chromatography with Buckyprep, and Buckyprep-D columns (Nacalai Tesque, Japan) as shown in Supplementary Figs 2–19. 0.5–2 mg of pure {Ln$_2$} compounds could be isolated in this work. The yield of {Ln$_2$} depends on the metal size: the yields of {Er$_2$}, {Ho$_2$}, and {Dy$_2$} are similar, the yield of {Tb$_2$} is ca twice lower, and the yield of {Gd$_2$} is the lowest in the series.

**Spectroscopic and electrochemical measurements**. Matrix-assisted laser desorption/ionization time-of-flight (MALDI-TOF) mass-spectra were measured with a Bruker autoflex mass-spectrometer with 1,1,4,4-tetraphenyl-1,3-butadiene as a matrix. EPR spectra of {Gd$_2$} solution in toluene were measured using cw-EPR spectrometer EMX Plus (Bruker), working in X-band and Q-band regions. The EPR spectra were fitted using Easyspin, a MATLAB toolbox[67]. UV-vis-NIR absorption spectra were measured in toluene solution at room temperature with a Shimadzu 3100 spectrophotometer. Raman spectra were recorded at 78 K on a T 64000 triple spectrometer (Jobin Yvon) using 656 nm excitation wavelength of a tunable dye laser Matisse 2 (Sirah Lasertechnik). IR spectra were measured at room temperature with Vertex 80 FTIR spectrometer (Bruker) equipped with a Hyperion microscope. For Raman and IR measurements, the {Ln$_2$} samples were drop-casted from toluene solution onto single-crystal KBr disks. NMR spectra were measured with an Avance 500 spectrometer (Bruker). Voltammetric experiments were performed with a sweep rate of 100 mV s$^{-1}$ in $o$-dichlorobenzene solution with TBABF$_4$ electrolyte salt in an oxygen-free glove box using potentiostat-galvanostat PARSTAT 4000 A. A three-electrode system with a platinum working and a counter electrode and a silver wire pseudo-reference electrode was used, potentials were calibrated by adding ferrocene as an internal standard in the end of each measurement.

**Single-crystal X-ray diffractometry**. Crystal growth of Dy$_2$@C$_{80}$-CH$_2$Ph•0.67 (CH$_3$Ph) was accomplished by layering hexane over a solution of {Dy$_2$} in toluene. Slow diffusion of two solutions resulted in formation of small black crystals (30 × 30 × 10 μm$^3$). X-ray diffraction data have been collected at 100, 130, 160, 190, 220, 250, 270, and 290 K on the BL14.3 beamline operated by the Joint Berlin MX

Laboratory at the BESSY II electron storage ring (Berlin-Adlershof, Germany)[68] using a MAR225 CCD detector, $\lambda = 0.89429$ Å. Processing diffraction data was done with XDSAPP2.0 suite[69]. The structure was solved by direct methods and refined using all data (based on F$^2$) by SHELX 2016[70]. Hydrogen atoms were located in a difference map, added geometrically, and refined with a riding model. Crystal data and data collection parameters are summarized in Supplementary Table 1

**Magnetometry**. DC magnetic measurements were performed using a Quantum Design VSM MPMS3 magnetometer. The samples were drop-casted from CS$_2$ solution into a standard powder sample holder. Long magnetization relaxation times of {Tb$_2$} were determined from the measurement of magnetization decay using a dc-SQUID. After the sample was magnetized to the saturation at 7 Tesla, the field was swept fast to zero or 0.3 T, and then the decay of magnetization was followed over hours and fitted with stretched exponential (Supplementary Figs 40, 41). Measurements of magnetic hysteresis curves of {Tb$_2$} were accomplished with a PPMS system equipped with a 14 T magnet. AC-magnetometry measurements were performed using Quantum Design MPMS XL magnetometer, Quantum Design VSM MPMS3 magnetometer, and PPMS system for the high-frequency range (0.5–10 kHz). See Supplementary Methods for further details.

**Calculations**. VASP code, version 5.0, was employed to optimize the molecular structures at the PBE-D level using PAW pseudopotentials[71]. Ab initio energies and wave functions of CF multiplets for the {LnY}$^-$ molecules have been calculated at the CASSCF/SO-RASSI level of theory using the quantum chemistry package MOLCAS 8.0[72]. The single ion LF-parameters were calculated based on ab initio data with the use of SINGLE_ANISO module[73]. Modelling of the magnetic properties was accomplished with the PHI program[60] and included powder-averaging. See Supplementary Methods for further details.

## Data availability

The X-ray crystallographic coordinates for the structures reported in this Article have been deposited at the Cambridge Crystallographic Data Centre (CCDC), under deposition numbers 1519744, 1851777, 1851778, 1851779, 1851780, 1851781, 1851782, 1851783. These data can be obtained free of charge from the CCDC via www.ccdc.cam.ac.uk/structures/? All other data supporting the findings of this study are available from the corresponding authors on request.

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

## Acknowledgements

The authors acknowledge funding from the European Union's Horizon 2020 research and innovation programme, European Research Council (grant agreement No 648295 to A.A.P.), and Marie Skłodowska-Curie action (grant agreement No. 748635 to S.M.A.); the Deutsche Forschungsgemeinschaft (grants PO 1602/4–1 and 1602/5-1 to A.A.P.), Slovak Research and Development Agency (grants APVV-15-0053, APVV-17-0513 to M.Z.), Slovak Scientific Grant Agency VEGA (grants 1/0416/17, 1/0466/18 to M.Z.), and the Swiss National Science Foundation (SNF Projects No. 206021_150784, No. 200021L_147201 to T.G.). R.R. and M. Richter acknowledge funding by the European Union (ERDF) and the Free State of Saxony via the ESF project 100231947 (Young Investigators Group Computer Simulations for Materials Design – CoSiMa). Diffraction data have been collected on BL14.3 at the BESSY II electron storage ring operated by the Helmholtz-Zentrum Berlin; we would particularly like to acknowledge the help and support of Manfred Weiss and his group members during the experiments at BESSY II. Computational resources were provided by the Center for High Performance Computing at the TU Dresden. We also thank L. Hozoi, J. van den Brink, G. Seifert, M.D. Kuzmin, and M. Boesler for helpful discussions and U. Nitzsche for technical support.

## Author contributions

F.L. and L.S. performed synthesis and X-ray diffraction studies, G.V. and D.S.K. studied magnetic properties with the help of K.N. and A.K. and under supervision of T.G., A.U.B.W., and B.B. E.P.R. studies were performed by M.Z. and M. Rosenkranz, M. Rosenkranz also did NMR measurements. S.S. and F.Z. performed optical spectroscopic characterization, N.A.S. and C.-H.C. did electrochemical measurements. S.M.A., R.R., and M. Richter performed DFT and ab initio calculations as well as the theoretical analysis. A.A.P. conceived and coordinated the project, simulated magnetic properties, and wrote the manuscript with contribution and discussion from all co-authors.

## Additional information

**Competing interests:** The authors declare no competing interests.

