## [Transparent Peer Review File · Nature Communications]

Reviewers' comments:

Reviewer #1 (Remarks to the Author):

The manuscript by Liu et al. presents the characterisation and magnetic properties of the Ln₂@C80-CH₂Ph series, featuring anisotropic Ln ions bound together with a single unpaired electron 'bond'. This results in strong magnetic exchange interactions and very interesting magnetic properties.

I think the work is of high interest to the readers of Nature Communications, but there are at present some major inconsistencies in the interpretation that must be fixed before it can be considered for publication.

The most significant is the modelling of the exchange interaction. Equations 2 and 3 describe the interaction of the bonding electron with the localised 4f moments (coupled into total angular momenta, J), as well as the local LF of each site - this is fine. However, they say: "To describe the lanthanide moment we use a pseudospin approach, in which ... Kramers doublets ... are treated as pseudospins $S = 1/2$ scaled with corresponding pseudospin g-tensors to match the size of the total spin.". Not only is this strange (why use a LF decomposition and then a pseudo-spin $S = 1/2$ model?), it is an inappropriate simplification in this case where the exchange couplings are large and are an appreciable fraction of the LF splittings (see Tables S5 - S7).

It also seems the authors are confused about their own model. When describing Equation 6, which is largely similar to Equations 2+3 and also contains the LF Hamiltonians for the Ln sites: "Since the relative energies of the first and second excited doublets, KD₂ and KD₃, are predicted to be in the range of 300–500 cm⁻¹ for Tb and 200–400 cm⁻¹ for Dy, only the lowest energy excited states contribute to the magnetic properties in the experimentally relevant temperature range. " - if a pseudospin $S = 1/2$ model is used, the LF is implicit and cannot be part of the Hamiltonian!

The authors should be using the full J,mJ basis for modelling their data, including the exchange. They should also be very careful using PHI, as I believe this makes some strange default assumptions when doing exchange coupling with lanthanides (the Hamiltonian is not simply $-2K(J.J)$) - this should be checked in the manual.

Further, when the authors use their pseudospin $S = 1/2$ model to compare to the U_{eff} values (which is an incorrect approach, based on the above), they seem to have done this incorrectly. They say "a more precise estimation is possible through modeling of the effective barriers of the Orbach process, U_{eff}. It can be done assuming that the low-energy LF excited states do not participate in the relaxation of {Tb₂} and {Dy₂}, while an efficient Orbach process involves the first exchange-excited state corresponding to flipping of one of the lanthanide spins under the Hamiltonian Eq. 6." Under the pseudospin approach with Hamiltonian $-2K(S.S)$ you get a fourfold ground state for $K > 0$, with a pair of doublets at K and 3K. $K = 55$ or 32 cm⁻¹ does not give first excited states at 556 and 426 cm⁻¹ (U_{eff} of Tb₂ and Dy₂ in cm⁻¹). The same problem comes in the analysis of Ho₂. They seem to be going down a better path in the SI with figures S64 and S65, where it seems they have included all LF states in the model, and look at the transition probability from the exchange model.

Other points:

- I really like the addition of paramagnetic NMR, and think the conclusions about TbY vs TbGd and Gd₂ are very interesting.

- what code for EPR simulations? Needs to be mentioned in text.

- The authors find an exponential temperature dependence of magnetic relaxation for Ho₂ at low

temperatures with a small Ueff barrier that is clearly not a 'normal' Orbach process. Sessoli and Yamashita ascribe such behaviour to a local vibrational mode (10.1021/jacs.8b06733), while Zheng and Chilton suggest it is a temperature dependent QTM. Can either of these models explain your data?

- The authors say the fits for Ho2 are worse than for Tb2 or Dy2. Actually, the fits look better than for Tb2!

- More exchange coupling problems. The authors briefly mention the non-collinearity of Ho2, and suggest using DM interactions. The non-collinearity arises from the local LF of each site, so why use DM? This would be included already if they get the exchange model correct using the LF and full J,mJ basis as discussed above. Besides, they show no results about this, so why is it even discussed?

- too much information in figure 4, this should be broken up.

- I suggest a figure directly comparing hysteresis and relaxation rates for Tb2 vs. TbGd and TbY

- electrochemistry should come before magnetism, characterising the nature of these interesting molecules

- the discussion puts a lot of emphasis on the non-collinearity of Ho axes being the origin of its poor SMM performance, however the LF states are VERY mixed - this is more likely the reason for its bad SMM performance, not the non-collinearity. If the authors get the exchange model right, they could do a quick test: B2O only for two Ho sites, however, rotate one a little bit and use the same exchange - does everything get stuffed up?

Reviewer #2 (Remarks to the Author):

This manuscript by Popov, Avdoshenko, Liu and their co-workers represents a detailed investigation toward a series of Ln-EMF with the formula $\text{Ln}_2@C_{80}(\text{CH}_2\text{Ph})$, including the preparation, structure, electrochemistry, magnetic properties and related theoretical calculations. Among the reported EMFs, the Tb2 species show high performance of magnetic blocking, with a gigantic coercive field and high blocking temperature. The electronic structure of EMFs together with their unique properties is an interesting topic and should be of interest to reader board of Nature Communications. However, there are several things confusing me. Thus, I recommend the publication of this paper in Nature Communications after some revisions as following:

(1) The checkcif report of the cifs alerts ADDSYM Detects New (Pseudo) Symmetry Element during the check process. I recommend the authors to check it carefully and if it is possible, fcf data should be added in the cifs.

(2) As I know, it is very hard to accumulate EMFs samples. Therefore, the magnetic measurement is very hard and usually has larger uncertainty comparing to the other coordination complexes. I suggest the authors adding the detailed information into the ESI, for example the methodology of sample preparation and data correction for magnetic measurements as well as the mass of each samples. The data looks so nice indicating that the amount of the sample should be quite much.

Reviewer #3 (Remarks to the Author):

This manuscript describes the synthesis and magnetic properties of a series of dimetallofullerenes containing lanthanide ions bonded by a single unpaired electron. The unpaired electron in these molecules facilitates strong ferromagnetic coupling, which in turn leads to magnetic blocking at high temperatures in the {Dy2} and {Tb2} analogues. The magnetic properties of these molecules

are quite impressive, and the analysis is thorough. I therefore recommend it for publication following minor revisions.

1) Since the submission of this paper, two manuscripts have been published that report 100-s blocking temperatures higher than $[DyCp^*ttt_2][B(C_6F_5)_4]$. These manuscripts should be cited, as they modify the claim that $\{Tb_2\}$ is the second highest 100-s blocking temperature yet reported for a single-molecule magnet. Note that this does not, however, diminish at all the importance of the results reported in this paper.

Guo, F.-S.; Day, B. M.; Chen, Y.-C.; Tong, M.-L.; Mansikkimäki, A.; Layfield, R. A. *Science*, 2018, DOI: 10.1126/science.aav0652.

McClain, K. R.; Gould, C. A.; Chakarawet, C.; Teat, S. J.; Groshens, T. J.; Long, J. R.; Harvey, B. G. *Chem. Sci.* 2018, DOI: 10.1039/C8SC03907K.

2) The manuscript would be improved by a discussion of the crystal field environment of the lanthanide ions. Based on the higher magnetic anisotropy of $\{Dy_2\}$ and $\{Tb_2\}$ as compared to $\{Ho_2\}$ and $\{Er_2\}$, it seems that an axial ligand field is imposed. Is this coming from interaction with the fullerene walls? Can this be rationalized through the crystal structure? And how strong is this interaction? This is an important question to answer, as both high anisotropy and strong magnetic exchange coupling are required for high-temperature magnetic blocking. If the crystal field splitting of a lanthanide ion's mJ states is quite low (i.e. low anisotropy), but strong exchange coupling is present, fast relaxation will likely be observed (see Vieru et. al. *Sci. Rep.* 2016, 6, 24046). I would expect this to be the case for these systems and am a bit surprised that the fullerene facilitates sufficiently high anisotropy in $\{Dy_2\}$ and $\{Tb_2\}$ to observe such high magnetic blocking temperatures, especially with the high calculated magnitude of the exchange coupling. While the magnetic properties are indisputable, a discussion of how anisotropy is set in these systems would improve the clarity of the manuscript.

3) Why is the exchange stronger in $\{Ho_2\}$ than in $\{Tb_2\}$ when it decreases from $\{Gd_2\}$ to $\{Tb_2\}$? This seems to counter the expected periodic trend (and that observed from $\{Gd_2\}$ to $\{Tb_2\}$). A discussion in this section would further clarify the manuscript.

We are thankful to reviewers for constructive and positive evaluation of our manuscript. Revised version of the manuscripts addresses all points raised by the reviewers, all changes are highlighted in yellow. Besides, the manuscript and supplementary information are edited to follow guidelines of the journal. Next pages present point-to-point response to reviewers' remarks.

Reviewers' comments:

Reviewer #1 (Remarks to the Author):

The manuscript by Liu et al. presents the characterisation and magnetic properties of the Ln₂@C80-CH₂Ph series, featuring anisotropic Ln ions bound together with a single unpaired electron 'bond'. This results in strong magnetic exchange interactions and very interesting magnetic properties.

I think the work is of high interest to the readers of Nature Communications, but there are at present some major inconsistencies in the interpretation that must be fixed before it can be considered for publication.

The most significant is the modelling of the exchange interaction. Equations 2 and 3 describe the interaction of the bonding electron with the localised 4f moments (coupled into total angular momenta, J), as well as the local LF of each site - this is fine. However, they say: "To describe the lanthanide moment we use a pseudospin approach, in which ... Kramers doublets ... are treated as pseudospins $S = 1/2$ scaled with corresponding pseudospin g-tensors to match the size of the total spin.". Not only is this strange (why use a LF decomposition and then a pseudo-spin $S = 1/2$ model?), it is an inappropriate simplification in this case where the exchange couplings are large and are an appreciable fraction of the LF splittings (see Tables S5 - S7).

It also seems the authors are confused about their own model. When describing Equation 6, which is largely similar to Equations 2+3 and also contains the LF Hamiltonians for the Ln sites: "Since the relative energies of the first and second excited doublets, KD2 and KD3, are predicted to be in the range of 300–500 cm⁻¹ for Tb and 200–400 cm⁻¹ for Dy, only the lowest energy excited states contribute to the magnetic properties in the experimentally relevant temperature range. " - if a pseudospin $S = 1/2$ model is used, the LF is implicit and cannot be part of the Hamiltonian!

The authors should be using the full J,mJ basis for modelling their data, including the exchange. They should also be very careful using PHI, as I believe this makes some strange default assumptions when doing exchange coupling with lanthanides (the Hamiltonian is not simply $-2K(J.J)$) - this should be checked in the manual.

We are very thankful to the Reviewer for these comments since they highlighted a confusion in the description of the theoretical model. We completely agree with the Reviewer that the molecules we study require the use of the full $|J, m_J\rangle$ basis, and this is exactly what we do in the manuscript. Thus, we believe that the criticism comes from a simple misunderstanding, caused by mentioning the pseudospin approach. In the revised manuscript we removed mentioning of the pseudospin approach and explicitly described that the full $|J, m_J\rangle$ basis is used.

However, we may note that the pseudospin description does not contradict this description. Ligand field splits the J levels of a lanthanide ion into separate $|J, m_J\rangle$ states, which for Kramers ions are two-fold degenerate and are known as Kramers doublets (KDs). In a pseudospin model, **each KD** is described as a pseudospin $S = 1/2$ **with a corresponding g-tensor**, which is (usually) highly anisotropic. E.g., for Dy^{3+} ion ($J = 15/2$) in the strongly axial LF, the ground state KD is $m_J = 15/2$, which in the pseudospin approach will be described as a $S=1/2$ with the diagonal g-tensor $(0, 0, 20)$. Likewise, the first excited state with $m_J=13/2$ is described by a pseudospin with $g=(0, 0, 18)$ and so on. Thus, if J states are described by a manifold of pseudospins with different g-tensors, equivalent description is obtained.

Further, when the authors use their pseudospin $S = 1/2$ model to compare to the U_{eff} values (which is an incorrect approach, based on the above), they seem to have done this incorrectly. They say "a more precise estimation is possible through modeling of the effective barriers of the Orbach process, U_{eff} . It can be done assuming that the low-energy LF excited states do not participate in the relaxation of $\{Tb_2\}$ and $\{Dy_2\}$, while an efficient Orbach process involves the first exchange-excited state corresponding to flipping of one of the lanthanide spins under the Hamiltonian Eq. 6." Under the pseudospin approach with Hamiltonian $-2K(S.S)$ you get a fourfold ground state for $K > 0$, with a pair of doublets at K and $3K$. $K = 55$ or 32 cm^{-1} does not give first excited states at 556 and 426 cm^{-1} (U_{eff} of Tb_2 and Dy_2 in cm^{-1}). The same problem comes in the analysis of Ho_2 . They seem to be going down a better path in the SI with figures S64 and S65, where it seems they have included all LF states in the model, and look at the transition probability from the exchange model.

As above, these is a misunderstanding caused by confusing description of the model in the original manuscript. We use the same approach for SI figures S64 and S65 and in the main text. For the three-center spin system, which we use to describe $\{Ln_2\}$ molecules, exchange term is written down as $-2K^{eff}\hat{s}(\hat{J}_1 + \hat{J}_2)$. Consider $\{Ln_2\}$ molecule, in which both Ln ion experience strong uniaxial ligand field (i.e. ground state has $J_z = \pm J$). The energy of the ground state would be then $-2JK^{eff}$, and the energy of the state, in which one of the Ln spins is flipped would be just 0. For $\{Tb_2\}$ with $J = 6$ and $K^{eff} = 55$ cm^{-1} , this would give the energy of the exchange excited state 660 cm^{-1} . If LF terms of the Hamiltonian are also taken into account, mixing of the LF and exchange excited states changes the energy of such a state to 556 cm^{-1} (see Supplementary Table 13 for exact composition of this state). This discussion is also added to the revised manuscript.

Other points:

- I really like the addition of paramagnetic NMR, and think the conclusions about TbY vs $TbGd$ and Gd_2 are very interesting.

- what code for EPR simulations? Needs to be mentioned in text.

EPR spectra were simulated with EasySpin, the reference is added to the manuscript.

- The authors find an exponential temperature dependence of magnetic relaxation for Ho₂ at low temperatures with a small U_{eff} barrier that is clearly not a 'normal' Orbach process. Sessoli and Yamashita ascribe such behaviour to a local vibrational mode (10.1021/jacs.8b06733), while Zheng and Chilton suggest it is a temperature dependent QTM. Can either of these models explain your data?

We are thankful to the Reviewer for putting these issues forward. Indeed, we do not think that it is a "normal" Orbach process. In the revised manuscript we discussed a possibility of the temperature dependence of the QTM due to the temperature-dependent phonon collision rate as suggested by Zheng and Chilton. As to a local vibrational mode, such possibilities have been discussed since early 1960s (Klemens, Phys. Rev. 1962, 125, 1795; Mills, Phys. Rev. 1966, 146, 336; etc). This mechanism might indeed explain Arrhenius behavior with unconventional parameters (i.e. unphysical for the Orbach mechanism), but cannot explain the field dependence observed for {Ho₂}. Therefore, we do not discuss it here.

- The authors say the fits for Ho₂ are worse than for Tb₂ or Dy₂. Actually, the fits look better than for Tb₂!

In the fitting of the magnetic data we used the followed strategy: χT curve measured for a given {Ln₂} compound was fitted for only one value of the magnetic field, 1 T, to determine K^{eff} . This K^{eff} was then used to calculate χT curves for other values of the magnetic field (3, 5, and 7 T) as well as to calculate magnetization curves at different temperatures. The agreement of the model and experiment is considered to be good, when a single K^{eff} value can give a good agreement for the whole set of experimental data. This remark is added to the Supplementary Methods.

For {Ho₂}, χT data are indeed fitted well for the magnetic field of 1 T, but when the determined K^{eff} constant is then used to model χT curves for other fields, considerable deviations between experiment and theory is found. Besides, computed and experimental magnetization curves for {Ho₂} deviate significantly. For {Tb₂}, the agreement between experiment and model remains reasonably good for all fields and for magnetization curves measured at different temperatures.

- More exchange coupling problems. The authors briefly mention the non-collinearity of Ho₂, and suggest using DM interactions. The non-collinearity arises from the local LF of each site, so why use DM? This would be included already if they get the exchange model correct using the LF and full J,mJ basis as discussed above. Besides, they show no results about this, so why is it even discussed?

The Hamiltonian (6) implies that the exchange interaction is described by an isotropic constant K^{eff} . However, we acknowledge that for non-collinear spins the use of isotropic exchange constant is an oversimplification, and one may have to use tensorial description, in which DM term appears naturally. We therefore mention this possibility, but not elaborate it further since the complexity of the situation exceeds the scope of this paper.

- too much information in figure 4, this should be broken up.

- I suggest a figure directly comparing hysteresis and relaxation rates for Tb₂ vs. TbGd and TbY

Following suggestion of the Reviewer, we (a) added {TbY} magnetization data to Figure 4 and (b) split the Figure 4 into two figures, one with the magnetic measurements and alignment of magnetic moments (new Fig. 4), and another one with relaxation times (new Fig. 5). Note that the addition of {TbGd} data is not reasonable at this moment. As we describe in the manuscript, the {TbGd} could not be separated from

$\{\text{Tb}_2\}$ and $\{\text{Gd}_2\}$, and hence the studied sample contained a mixture of all three compounds. Hence, we use the data for this sample only when the contribution of $\{\text{TbGd}\}$ is clearly different from that of $\{\text{Tb}_2\}$ and $\{\text{Gd}_2\}$, such as in ^1H NMR spectra and in the blocking temperature measurements. In magnetization curves the contribution of $\{\text{TbGd}\}$ cannot be clearly discerned from that of $\{\text{Gd}_2\}$ and $\{\text{Tb}_2\}$, and we prefer not to use such data in the paper.

- electrochemistry should come before magnetism, characterising the nature of these interesting molecules

The proper place for the electrochemical part was also a matter of concern for us during writing the manuscript. We agree that the electrochemical part alone would better come before magnetism. However, discussion of the paramagnetic NMR data for anions, the influence of the charge transfer on the exchange interactions, and an outlook about the possible transport properties, which logically follow from the results of the electrochemical measurements, are also based on the results of the magnetic measurements. Therefore, we decided to place the whole discussion of the electron transfer properties after the magnetism.

- the discussion puts a lot of emphasis on the non-collinearity of Ho axes being the origin of its poor SMM performance, however the LF states are VERY mixed - this is more likely the reason for its bad SMM performance, not the non-collinearity. If the authors get the exchange model right, they could do a quick test: B20 only for two Ho sites, however, rotate one a little bit and use the same exchange - does everything get stuffed up?

We are thankful to the Reviewer for this useful comment. In fact, our current understanding of the $\{\text{Ho}_2\}$ system does not allow us to favor one particular reason. In the revised version of the manuscript, both mixing of LF states and non-collinearity are mentioned as possible reasons for the poor SMM performance of $\{\text{Ho}_2\}$.

Reviewer #2 (Remarks to the Author):

This manuscript by Popov, Avdoshenko, Liu and their co-workers represents a detailed investigation toward a series of Ln-EMF with the formula $\text{Ln}_2@C_{80}(\text{CH}_2\text{Ph})$, including the preparation, structure, electrochemistry, magnetic properties and related theoretical calculations. Among the reported EMFs, the Tb_2 species show high performance of magnetic blocking, with a gigantic coercive field and high blocking temperature. The electronic structure of EMFs together with their unique properties is an interesting topic and should be of interest to reader board of Nature Communications. However, there are several things confusing me. Thus, I recommend the publication of this paper in Nature Communications after some revisions as following:

(1) The checkcif report of the cifs alerts ADDSYM Detects New (Pseudo) Symmetry Element during the check process. I recommend the authors to check it carefully and if it is possible, fcf data should be added in the cifs.

Following Reviewer's recommendation, we checked the possible pseudo symmetry with ADDSYM search, but no obvious space group change is needed/suggested. We believe the space group P-1 is proper in this case. Cif-file with fcf data is uploaded as supplementary information; besides, cif-files for each temperature are deposited to CCDC (this information is also added to the manuscript as Data availability statement)

(2) As I know, it is very hard to accumulate EMFs samples. Therefore, the magnetic measurement is very hard and usually has larger uncertainty comparing to the other coordination complexes. I suggest the authors adding the detailed information into the ESI, for example the methodology of sample preparation and data correction for magnetic measurements as well as the mass of each samples. The data looks so nice indicating that the amount of the sample. should be quite much.

We are thankful to the Reviewer for this comment. Accumulation of EMF samples is indeed not a simple task. As we described in SI, the amount of samples available for the measurements ranged from 0.5 mg for {Gd₂} to 2 mg {Ho₂} and {Er₂}. Since magnetic moments of these molecules are rather large, good-quality DC measurements of {Ln₂} compounds with VSM-option of the MPMS-3 magnetometer require ca 0.1 mg. Certainly, to ensure low background signal it is important to use well-handled sample rod and high quality quartz sample holders. AC measurements require at least 0.5 mg, and good-quality curves are obtained when the sample amount exceeds 1 mg. Besides, we used three different magnetometers for AC measurements to have optimal signal-to-noise ratio in different frequency ranges. With these precautions in mind, the sample amount was sufficient for good-quality measurements without any specific adjustments. More detailed description than in the original manuscript is now provided in the Methods section.

Reviewer #3 (Remarks to the Author):

This manuscript describes the synthesis and magnetic properties of a series of dimetallofullerenes containing lanthanide ions bonded by a single unpaired electron. The unpaired electron in these molecules facilitates strong ferromagnetic coupling, which in turn leads to magnetic blocking at high temperatures in the {Dy₂} and {Tb₂} analogues. The magnetic properties of these molecules are quite impressive, and the analysis is thorough. I therefore recommend it for publication following minor revisions.

1) Since the submission of this paper, two manuscripts have been published that report 100-s blocking temperatures higher than [DyCp₂ttt₂][B(C₆F₅)₄]. These manuscripts should be cited, as they modify the claim that {Tb₂} is the second highest 100-s blocking temperature yet reported for a single-molecule magnet. Note that this does not, however, diminish at all the importance of the results reported in this paper.

Guo, F.-S.; Day, B. M.; Chen, Y.-C.; Tong, M.-L.; Mansikkimäki, A.; Layfield, R. A. *Science*, 2018, DOI: 10.1126/science.aav0652.

McClain, K. R.; Gould, C. A.; Chakarawet, C.; Teat, S. J.; Groshens, T. J.; Long, J. R.; Harvey, B. G. Chem. Sci. 2018, DOI: 10.1039/C8SC03907K.

The references were added and the text of the manuscript was updated to take these works into account.

2) The manuscript would be improved by a discussion of the crystal field environment of the lanthanide ions. Based on the higher magnetic anisotropy of {Dy²⁺} and {Tb²⁺} as compared to {Ho²⁺} and {Er²⁺}, it seems that an axial ligand field is imposed. Is this coming from interaction with the fullerene walls? Can this be rationalized through the crystal structure? And how strong is this interaction? This is an important question to answer, as both high anisotropy and strong magnetic exchange coupling are required for high-temperature magnetic blocking. If the crystal field splitting of a lanthanide ion's m_J states is quite low (i.e. low anisotropy), but strong exchange coupling is present, fast relaxation will likely be observed (see Vieru et. al. Sci. Rep. 2016, 6, 24046). I would expect this to be the case for these systems and am a bit surprised that the fullerene facilitates sufficiently high anisotropy in {Dy²⁺} and {Tb²⁺} to observe such high magnetic blocking temperatures, especially with the high calculated magnitude of the exchange coupling. While the magnetic properties are indisputable, a discussion of how anisotropy is set in these systems would improve the clarity of the manuscript.

As follows from ab initio calculations (Supplementary Tables 8), single-ion magnetic anisotropy in {Ln₂} is rather high. For instance, the total LF splitting for Tb ions is ca 1000 cm⁻¹, and the energy of the first excited KD state is ca 260 cm⁻¹. Ligand field is indeed highly axial, resulting in high-spin ground states of Ising type.

The axially of the ligand field in {Ln₂} may have several reasons. First, metal atoms transfer their valence electrons to the fullerene cage, resulting in accumulation of the negative charge on carbon atoms coordinated to the endohedral lanthanide ion. Note that this interaction also has considerable covalent contribution via overlap of π-electron density of the fullerene with vacant d-orbital of the lanthanide. Next, covalent Ln-Ln bonding results in a concentration of the electron density between two Ln ions. In Ref. 38 we used a simple point-charge model to show that even relatively small negative charge located between two lanthanide ions may induce rather high axial magnetic anisotropy. Thus, metal-metal bond is important not only for exchange interactions, but also to support the axial ligand field. Finally, lanthanide ions in EMFs have no “equatorial” ligands – the situation which also facilitates imposing of the axial ligand field. These factors are discussed in Supplementary Note 3.

Variation of the anisotropy along the lanthanide series, in particular high easy-axis anisotropy for Tb and Dy, moderate anisotropy for Ho, and easy-plane anisotropy for Er are discussed in Supplementary Note 4.

3) Why is the exchange stronger in {Ho²⁺} than in {Tb²⁺} when it decreases from {Gd²⁺} to {Tb²⁺}? This seems to counter the expected periodic trend (and that observed from {Gd²⁺} to {Tb²⁺}). A discussion in this section would further clarify the manuscript.

According to the fits to the experimental data, the K^{eff} value is decreasing from 160 cm^{-1} for $\{\text{Gd}_2\}$ to 55 cm^{-1} for $\{\text{Tb}_2\}$ to 32 cm^{-1} for $\{\text{Dy}_2\}$, which follows the intuitive expectations. At the same time, the optimal value for $\{\text{Ho}_2\}$, 40 cm^{-1} , is indeed higher than that of $\{\text{Dy}_2\}$. However, we should note that the validity of the Hamiltonian 6 for the $\{\text{Ho}_2\}$ system is questionable, and the determined K^{eff} parameter is not as reliable as that for $\{\text{Tb}_2\}$ and for $\{\text{Dy}_2\}$. Since exchange interaction in $\{\text{Ho}_2\}$ most probably have more complex nature and may require a tensorial description of the exchange interactions, we prefer not to make far-going conclusions on the trend in K^{eff} values along the lanthanide row. For the same reason, the K^{eff} value of 20 cm^{-1} estimated for $\{\text{Er}_2\}$ using Hamiltonian 6 is also not discussed further.

REVIEWERS' COMMENTS:

Reviewer #1 (Remarks to the Author):

Generally I am satisfied with the replies from the authors. While they have made some minor additions to the manuscript, I believe they still need to further clarify their statements so as not to confuse readers.

They have provided significant information in response to the referee comments, but mostly these responses and clarifications have not been incorporated into the manuscript. The authors should reflect on the points they have been asked to clarify with the referees, and make sure these aspects are clarified in full in the text.

In their response, the authors have clarified that they have used the full J, m_J basis for both Ln sites - this is good. However, they go on to defend the pseudo spin model: "Thus, if J states are described by a manifold of pseudospins with different g-tensors, equivalent description is obtained." - this comment is incorrect. It may be a good approximation for the low-field magnetic properties of a single ion, but it is incorrect in applied fields (because the magnetic field operator will not be able to mix the states correctly) and when coupling is involved (because the exchange operators will be mixing $m_s \pm 1/2$ states, and not the actual m_J states)!! I would request the authors confirm to the editors, before the MS is published, that they have not used this approach because it is simply wrong.

Reviewer #3 (Remarks to the Author):

The revised manuscript addresses all of the points raised in the initial review, so I recommend it for publication.

REVIEWERS' COMMENTS:

Reviewer #1 (Remarks to the Author):

Generally I am satisfied with the replies from the authors. While they have made some minor additions to the manuscript, I believe they still need to further clarify their statements so as not to confuse readers.

They have provided significant information in response to the referee comments, but mostly these responses and clarifications have not been incorporated into the manuscript. The authors should reflect on the points they have been asked to clarify with the referees, and make sure these aspects are clarified in full in the text.

In their response, the authors have clarified that they have used the full J, m_J basis for both L_n sites - this is good. However, they go on to defend the pseudo spin model: "Thus, if J states are described by a manifold of pseudospins with different g -tensors, equivalent description is obtained." - this comment is incorrect. It may be a good approximation for the low-field magnetic properties of a single ion, but it is incorrect in applied fields (because the magnetic field operator will not be able to mix the states correctly) and when coupling is involved (because the exchange operators will be mixing $m_s \pm 1/2$ states, and not the actual m_J states)!! I would request the authors confirm to the editors, before the MS is published, that they have not used this approach because it is simply wrong.

In the revision of the manuscript following referees' requests, we had to keep the balance with the size of the manuscript. Therefore lengthy explanations were added to SI, but we did our best to respond to all critical comments in the manuscript as well. Thanks to Reviewer 1, the most critical issue of the manuscript, description of the modelling of magnetic properties, has been corrected. We changed the confusing statement in the initial version (statement about the pseudospin) to the correct description, stating that the full basis is used. Following addition request of the reviewer, we would like to confirm to the Editors that pseudospin approach was not used in the modelling of magnetic data.